## TECHNIQUES AND RESOURCES

# Genomic resources for the scuttle fly *Megaselia abdita*: a model organism for comparative developmental studies in flies

Ayse Tenger-Trolander[1,*], Ezra Amiri[1], Valentino Gantz[2,3], Chun Wai Kwan[1,4], Himanshi Yadav[5], Sheri A. Sanders[6] and Urs Schmidt-Ott[1,*]

## ABSTRACT

The order Diptera (true flies) holds promise as a model taxon in evolutionary developmental biology due to the inclusion of the model organism *Drosophila melanogaster* and the ability to cost-effectively rear many species in laboratories. One of these dipteran species, the scuttle fly *Megaselia abdita* (Phoridae), has been used in evolutionary developmental biology for 30 years and is an excellent phylogenetic intermediate between fruit flies and mosquitoes, but remains underdeveloped in genomic resources. Here, we present a *de novo* chromosome-level assembly and annotation of *M. abdita* and transcriptomes of nine embryonic and four post-embryonic stages. We also compare nine stage-matched embryonic transcriptomes between *M. abdita* and *D. melanogaster*. Our analysis of these resources reveals extensive chromosomal synteny with *D. melanogaster*; 24 orphan genes with embryo-specific expression, including a novel F-box LRR gene in *M. abdita*; and conserved and diverged features of gene expression dynamics between *M. abdita* and *D. melanogaster*. Collectively, our results provide a new resource for studying the diversification of developmental processes in flies.

KEY WORDS: Genome assembly, Genome annotation, Transcriptome, Non-traditional model organism, Evolutionary development, Synteny analysis, Orphan genes

## INTRODUCTION

Comparing related species is a powerful approach to understanding how mechanisms of development evolve and diversify. Their natural diversity and their phylogenetic history can aid in revealing core principles and inform our understanding of evolutionary transitions. Additionally, careful comparisons of developmental mechanisms between multiple species in 'model taxa' are crucial for determining the directionality of change and identifying mutations responsible for important evolutionary shifts in developmental processes. Such mutations may not necessarily be adaptive, and their significance as drivers of evolutionary change might be overlooked, given that complex developmental gene networks can enhance a population's permissiveness for the passive fixation of mutational variants that open novel paths of adaptive evolution (Kimura and Ohta, 1974; Lynch, 2007a,b).

While it is impractical to adapt a large set of closely related vertebrate model organisms for laboratory studies, insects, in particular Diptera (true flies), offer a cost-effective alternative (Grimaldi and Engel, 2005; Schmidt-Ott and Lynch, 2016; Wiegmann et al., 2011). Flies are particularly appealing for the comparative study of developmental mechanisms because they include a leading model organism in developmental biology, *Drosophila melanogaster*, and many more species that are relatively easily cultured in laboratories. Developmental biologists have introduced several new dipteran model organisms in recent years, including the humpbacked fly *Megaselia abdita*, which has been particularly useful for studying the evolution of developmental mechanisms in dipteran embryos because of its technical advantages and phylogenetic position (Rafiqi et al., 2011). It belongs to the large family Phoridae, also known as scuttle flies (Disney, 1994; Li et al., 2025), and represents a lineage that separated from the *Drosophila* lineage ∼145 million years ago at the beginning of the Cyclorrhapha radiation, roughly 100 million years into the dipteran radiation (Grimaldi and Engel, 2005). Developmental biologists started using *M. abdita* as an experimental system in the 1990s to study the evolution of axial pattern formation (Bullock et al., 2004; Crombach and Jaeger, 2012; Crombach et al., 2016; Liu et al., 2018; Rohr et al., 1999; Schmidt-Ott et al., 1994; Stauber et al., 1999, 2000, 2002; Wotton et al., 2015a,b,c; Yoder and Carroll, 2006), and subsequently the evolution of extraembryonic tissue (Caroti et al., 2018; Fraire-Zamora et al., 2018; Horn et al., 2015; Kwan et al., 2016; Rafiqi et al., 2008, 2010, 2012; Schmidt-Ott and Kwan, 2022; Stauber et al., 1999; Wotton, 2014; Wotton et al., 2014) and other aspects of embryo development (Caroti et al., 2015; Dey et al., 2025; Tanaka et al., 2015; Vicoso and Bachtrog, 2015). However, the limited availability of genomic resources in *M. abdita* (Jiménez-Guri et al., 2013; Vicoso and Bachtrog, 2015) and the Phoridae in general (Feng et al., 2020; Rasmussen and Noor, 2009; Zhong et al., 2016) has limited the potential of this model organism by precluding genome-wide and epigenetic experimental approaches (Fig. 1).

Here, we provide a *de novo* assembled and annotated chromosome-level genome for *M. abdita*, alongside stage-specific transcriptomes across its life cycle. These resources provide an excellent basis for genome-wide and epigenetic experimental approaches for an understudied but phylogenetically important branch of the Diptera. They also establish synteny relationships with the chromosomes of *D. melanogaster*, highlight conserved and divergent features of Hox gene clusters, and provide insights into embryonic gene expression

[1]University of Chicago, Department of Organismal Biology and Anatomy, 1027 East 57th Street, Chicago, IL 60637, USA. [2]Section of Cell and Developmental Biology, University of California San Diego, La Jolla, CA 92093, USA. [3]Pattern Biosciences, 681 Gateway Blvd, South San Francisco, CA 94080, USA. [4]Laboratory for Epithelial Morphogenesis, RIKEN Center for Biosystems Dynamics Research, 2-2-3 Minatojima-minamimachi, Chuo-ku, Kobe, Hyogo 650-0047, Japan. [5]University of Chicago, Research Computing Center, 6054 S Drexel Ave, Office 354, Chicago, IL 60637, USA. [6]Notre Dame University, 252 Galvin Life Science Center/Freimann Life Science Center, Notre Dame, IN 46556, USA.

*Authors for correspondence (ayse.trolander@gmail.com; uschmid@uchicago.edu)

A.T.-T., 0000-0001-8319-0273; E.A., 0000-0002-6241-6656; V.G., 0000-0003-2453-0711; C.W.K., 0000-0003-3974-2979; H.Y., 0009-0007-3760-9564; S.A.S., 0000-0003-2969-2541; U.S.-O., 0000-0002-1351-9472

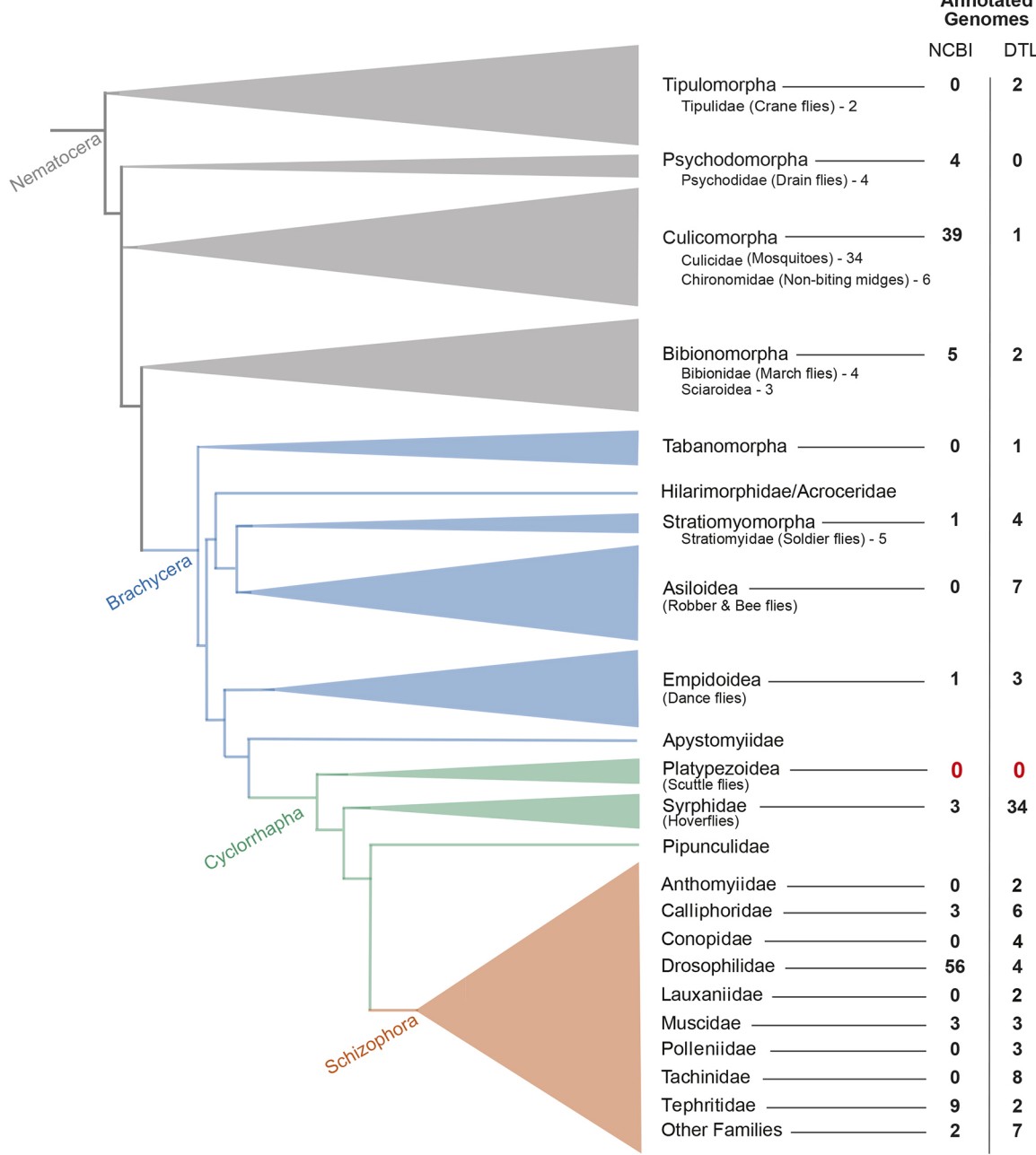

**Fig. 1. Phylogeny of Diptera with available genome annotations across lineages.** The phylogeny is based on Wiegmann et al. (2011) and Jiménez-Guri et al. (2013). The width of the base of each triangle reflects species richness within each group. The number of annotated genomes available for each clade in the genome database at National Center for Biotechnology Information (NCBI) or in the Darwin Tree of Life (DTL) project that have been uploaded as of February 4th 2025 are indicated. The red zeros highlight the phylogenetic position of *Megaselia abdita*.

dynamics and orphan genes. Collectively, our results will help to establish dipterans as a model taxon for studying the evolution of developmental mechanisms from gene regulation to neural networks and behavior.

## RESULTS AND DISCUSSION
### Genome assembly
We generated a *de novo* reference genome for *Megaselia abdita* using combined long read and chromatin conformation capture methods obtained from several hundred embryos of a 10-generation inbred line. Long-read, high-fidelity sequences were generated by PacBio (HiFi PacBio reads), and chromatin conformation capture sequences were generated by Dovetail Genomics' Omni-C method. After removing 33 scaffolds identified as contamination, the initial draft assembly spanned 592.8 megabases (Mb) contained in 89 scaffolds with an N50 of 12.6 Mb. Using HiRise, which is software designed to scaffold genome assemblies with proximity ligation data (Putnam et al., 2016), the draft assembly was refined using the Omni-C reads (Fig. S1).

The final assembly is 592.8 Mb contained in 15 scaffolds, with an N50 of 212.8 Mb (Table 1). The genome size is comparable to a previous estimate of 562.7 Mb based on flow cytometry data (Picard et al., 2012). We estimated heterozygosity at 0-0.24% (Fig. S2A). We masked 67.9% of the genome, constituting

**Table 1. *M. abdita* reference assembly statistics and quality metrics**

| Reference assembly statistics | |
|---|---|
| Total length (bp) | 592,824,975 |
| Number of scaffolds | 15 |
| Scaffold N50 (bp) | 212,802,314 |
| Scaffold L50 | 2 |
| Number of Ns per 100 kb | 1.32 |
| GC (%) | 29.74% |
| Masked (%) | 67.90% |
| Eukaryotic BUSCO's recovered (%) | 99% |

repetitive elements (Fig. S2B). The largest three scaffolds correspond to the three chromosomes of *M. abdita* (Table S1). The remaining 12 scaffolds contained repetitive sequences and could not be assembled into chromosome-level scaffolds.

### Genome annotation

We annotated the reference genome of *M. abdita* using evidence from RNA-Seq transcripts and protein sequence databases, and a robust genome annotation pipeline (Fig. S3). The process involved mapping RNA-Seq data and assembling the transcriptome, mapping protein sequences to the reference genome, and generating gene models using a variety of prediction software. We then created consensus gene models using EVidenceModeler, updated the models to include untranslated regions (UTRs) and alternative isoforms, and filtered out transposable element models before functionally annotating the genes with eggNOG-mapper (Haas et al., 2008; Huerta-Cepas et al., 2019; Cantalapiedra et al., 2021). In the Materials and Methods section, we provide a detailed walkthrough of this pipeline. In addition, we are working to develop a portable and reproducible pipeline using Nextflow, which allows users to publish or access bioinformatic pipelines (see the section 'Data and resource availability').

The genome of *M. abdita* contains 11,934 protein-coding genes (Table 2). We assessed the quality of the annotation with Benchmarking Universal Single-Copy Orthologs (BUSCO) (Simão et al., 2015; Manni et al., 2021). BUSCO evaluates annotation completeness by looking for the presence of highly conserved 'single-copy orthologs' across specific taxonomic groups. For example, the BUSCO Eukaryota database expects 255 orthologs, while the Diptera database expects 3285. For the *M. abdita* genome annotation, we found 93% eukaryotic and 88% dipteran 'complete single copy orthologs', indicating a high-quality annotation (Table 2).

**Table 2. Genome annotation statistics (top) and quality metrics (bottom) for *M. abdita***

| Annotation statistics | |
|---|---|
| Number of coding genes | 11,934 |
| Number of mRNAs | 20,560 |
| Number of mRNAs with 3′ & 5′ UTR | 17,607 |
| Mean mRNAs/gene | 1.7 |
| Mean gene length (bp) | 19,478 |
| Complete universal orthologs recovered in annotation by lineage-specific database (BUSCOs) | |
| Eukaryota (*n*=255) | 93.3% |
| Metazoa (*n*=954) | 90.0% |
| Insecta (*n*=1367) | 91.6% |
| Endopterygota (*n*=2124) | 90.6% |
| Diptera (*n*=3285) | 88.0% |

The quality metrics include the percentage of complete universal orthologs identified in the annotation across five BUSCO datasets. The number of genes in each lineage-specific BUSCO database is shown in parentheses.

The genome size of *M. abdita* is significantly larger than that of *D. melanogaster* (592.8 Mb versus 139.5 Mb). We also found the mean gene length of protein-coding genes in *M. abdita* to be significantly longer than that of *D. melanogaster*, ~19 kilobases (kb) versus ~6.9 kb, respectively. However, the lengths of transcripts and exons are remarkably consistent between the two species, indicating that the larger genome size of *M. abdita* is partially attributable to longer introns (Table S2).

### Genome browser and genomic analysis tools

To improve accessibility and usability of the genomic data hosted on NCBI, we developed a genome browser ecosystem centered on JBrowse2 (Diesh et al., 2023). This ecosystem is available as a cloud image on NSF's Jetstream2 platform (image Megaselia abdita Genome Resources, Hancock et al., 2021; Boerner et al., 2023), with a portable Docker image available on GitHub (https://github.com/kallistaconsulting/genomic_resources_megaselia). The browser integrates tools for comprehensive genomic analysis, including BLAST search (SequenceServer2.0, Priyam et al., 2019), CRISPR guide RNA design (modified crisprDesigner, Beeber and Chain, 2020), differential gene expression (DGE) analysis via R Shiny (freecount, Brooks et al., 2024) and synteny mapping (ShinySyn, Xiao and Lam, 2022). Future updates will include expanded sgRNA profiling, Docker support for deployment on commercial cloud platforms and continuing optimization of workflows, ensuring this resource remains a powerful and accessible tool for genomic research and education.

### Synteny analysis reveals significant collinearity between *M. abdita* and *D. melanogaster* genomes, including HOX gene cluster arrangement

Analysis of chromosomal synteny can aid in the identification of orthologs and highlight genomic regions with conserved regulatory potential. To investigate genome-wide synteny between *M. abdita* and *D. melanogaster*, we performed a synteny analysis, identifying significant collinearity between their genomes, including the arrangement of the split HOX gene cluster. We compared the synteny and collinearity of *M. abdita* scaffolds with the *D. melanogaster* genome and identified 387 collinear blocks encompassing 2396 genes (Fig. 2A). Large portions of *D. melanogaster* chromosomes are syntenic with single scaffolds in *M. abdita*. Scaffold 1 of *M. abdita* largely corresponds to chromosome arm 3L, chromosome 4 and chromosome arm 2R of *D. melanogaster*, while scaffold 2 aligns with chromosome arm 2L and the distal region of chromosome arm 3R. Scaffold 3 aligns with the proximal region of chromosome arm 3R and the X chromosome. The arrangement of the HOX genes in *M. abdita* mirrored that of *D. melanogaster*, with distinct Antennapedia and Ultrabithorax complexes separated by 53,306 kb in *M. abdita* (9978 kb in *D. melanogaster*). Both complexes are located on scaffold 2 of the genome of *M. abdita* and maintain the same gene order seen in *D. melanogaster*, including the cuticle gene complex (Fig. 2B). Additionally, the genes *zerknüllt* (*zen*) and *amalgam* have undergone duplication in *M. abdita* (Fig. 2B). *zen* has experienced many duplications in Diptera (Mulhair and Holland, 2024). Since only the ~60 amino acid homeodomain of *zen* is comparable between species, it is difficult to establish the relatedness of *M. abdita*'s zen-like (*Mab-zen-like*) to other Zen genes and duplications by sequence alone.

*M. abdita*'s *zen* gene (*Mab-zen*) is expressed in the serosa and has been characterized in detail (Caroti et al., 2018; Kwan et al., 2016; Rafiqi et al., 2008, 2010, 2012; Stauber et al., 1999). Consistent with these studies, we detected *Mab-zen* transcripts in a time series

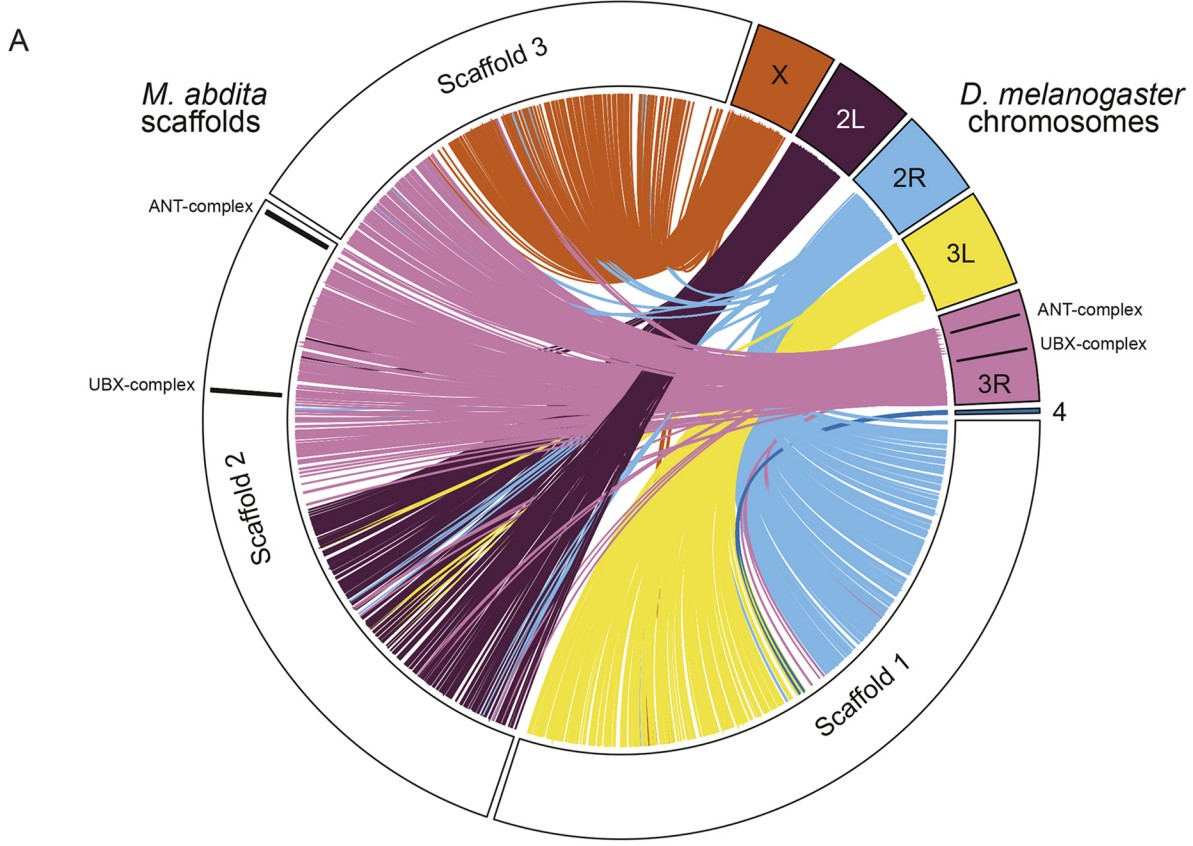

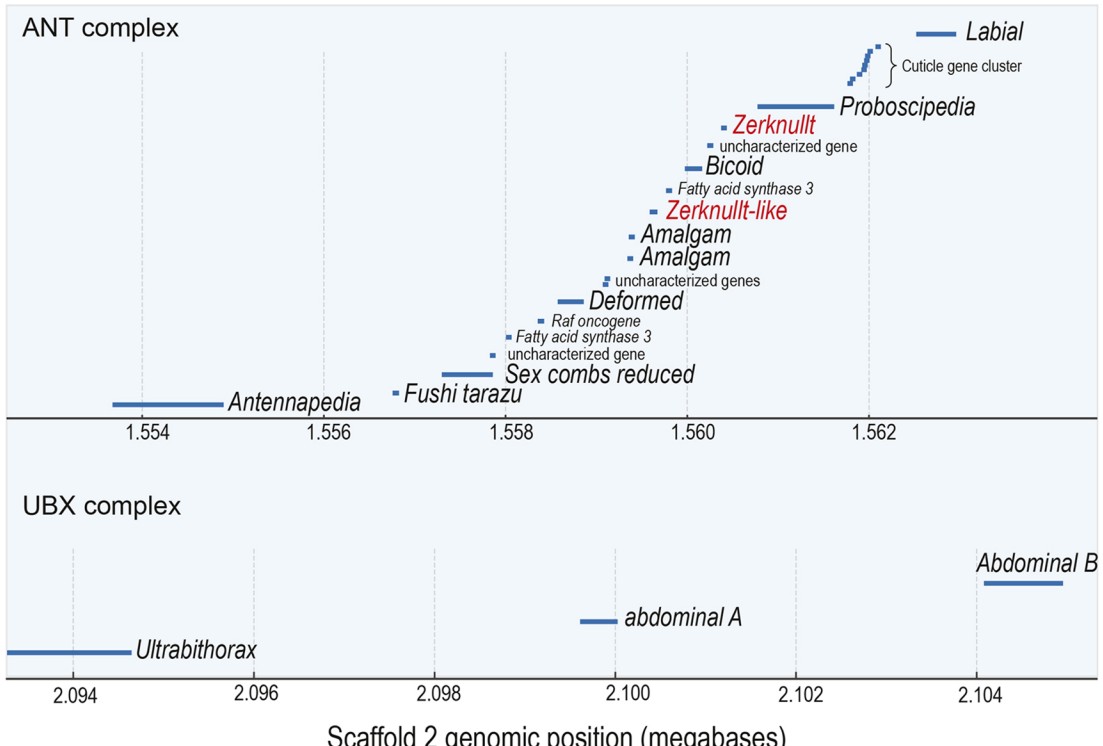

**Fig. 2. Synteny between *D. melanogaster* and *M. abdita*, and Hox gene organization in *M. abdita*.** (A) Synteny between *D. melanogaster* chromosomes and *M. abdita* scaffolds. Lines represent groups of collinear genes, connecting their positions between *D. melanogaster* chromosomes and *M. abdita* chromosome-sized scaffolds. Colors correspond to *D. melanogaster* chromosomes. The ANT and UBX Hox gene complexes are highlighted by black lines. (B) Visualization of *M. abdita* Hox gene clusters. The Antennapedia (top) and Ultrabithorax (bottom) complexes are both located on Scaffold 2. Each blue line indicates the genomic position and length of a gene. *Mab-Zerknullt* (*Mab-zen*) and *Mab-Zerknullt-like* (*Mab-zen-like*) are highlighted in red.

of single embryo transcriptomes (described in the next section) from stages 5 to 9, coinciding with the time when the serosa is specified but not at stage 10, and at very low levels at stages 12 and 13, when the completed serosa is maintained (Fig. S4A). In contrast, the newly identified *zen-like* gene was expressed only at stage 5 (Fig. S4A). To validate these RNA-Seq patterns in the absence of biological replicates, we performed RNA-FISH HCR across an embryonic time series (Fig. S4B). Staining confirmed the expression of *Mab-zen* during cellularization and germband extension, and the stage-specific expression of *Mab-zen-like* at stage 5 (Fig. S4B).

### Major transition in transcriptional expression profile during germband retraction

The extent to which gene expression is similar or different between stages and species provides a basis for identifying developmental windows of accelerated change, heterochronic shifts and evolutionary divergence. To identify conserved features and differences between embryonic stages of *M. abdita* and *D. melanogaster*, we performed RNA-seq on single embryos from stages 1, 5, 8, 9, 10, 12, 13, 15, 16 and 17 for both species (Fig. S5). We staged embryos based on morphology, corresponding to established staging schemes for each species (Campos-Ortega and Hartenstein, 1997; Wotton et al., 2014). We later excluded the *M. abdita* stage 16 embryo due to failed sequencing. Despite similar embryo sizes, and uniform library preparation and sequencing conditions, *M. abdita* embryos had roughly twice the number of reads per embryo compared to *D. melanogaster* (Fig. S6A), though the total number of genes with reads assigned was comparable between species with 10,373 in *D. melanogaster* and 9999 in *M. abdita*. The number of genes expressed was also similar across embryonic stages between species (Fig. S6B). The number of reads mapping to genes doubled in *M. abdita* (8941 versus 3863 reads/gene), which was consistent with its 2× read count. We used normalized read counts (transcripts per million or TPM) to account for the global expression level differences, sample-to-sample variation, and differences in transcript length between genes. Our RNA-Seq data showed a major shift in transcriptional expression during germband retraction in both *M. abdita* and *D. melanogaster*. A multidimensional scaling (MDS) plot, which plots samples based on the similarity of their top 500 most differentially expressed genes, revealed clustering of embryo transcriptomes before and after germband retraction (Fig. 3A).

To more closely look at the gene networks in both species, we identified the differentially expressed genes (DEGs) between developmental stages. Owing to the absence of biological replicates for each embryo, we used a k-mean clustering approach to group samples and calculated a global dispersion estimate ($\sigma$=0.36 for both *M. abdita* and *D. melanogaster*). Both species have high dispersion estimates but variance in expression was very similar in both datasets (Figs S7 and S8). The clustering was performed to estimate a reasonable global dispersion parameter for the datasets and is a conservative approach when using traditional hypothesis testing to identify significant DEGs. We used a threshold of fold differences ±3 and $P<0.001$.

We performed pairwise comparisons of each sequential stage: the zygote, containing maternal-deposited transcripts (stage 1); cellularization, where maternal to zygotic transition (MZT) of gene expression occurs (stage 5); gastrulation, germband extension and retraction (stages 8-12); dorsal closure (stages 13-16); and the final embryonic stage (stage 17) using the estimated global dispersion. We detected 1567 and 2398 DEGs in *M. abdita* and *D. melanogaster*, respectively (Fig. 3B). *M. abdita* exhibited more dynamic gene expression than *D. melanogaster* between

cellularization and gastrulation (stages 5-8), with 418 DEGs in *M. abdita* versus 123 in *D. melanogaster* (Fig. 3B). Between stages 8 and 12, corresponding with the period of germband extension and germband retraction, there were markedly fewer DEGs in both species than between stages 12 (germband retraction) and 13 (beginning of dorsal closure), when both species showed a strikingly dynamic shift in gene expression (Fig. 3B), albeit to a different extent. Between stages 12 and 13, *D. melanogaster* had 1463 DEGs compared to 564 DEGs in *M. abdita* (Fig. 3B), indicating sharper increases and decreases in gene expression with the onset of dorsal closure in *D. melanogaster* than in *M. abdita*. The increased turning on and off of genes between stages 12 and 13 in both species also coincides with the phylotypic stage of development when evolutionary divergence between species is most constrained (Sander, 1983; Kalinka et al., 2010), suggesting that dynamic changes in gene expression impose evolutionary constraints on gene regulatory networks.

### GO term enrichment identifies conserved and divergent embryonic processes

We performed soft clustering with the R package Mfuzz on our RNA-Seq dataset, which allowed us to look at gene expression patterns in our time course dataset without estimating variance (estimated with biological replicates, which we lack). Mfuzz has been widely applied for developmental time-series RNA-seq analyses across diverse systems, including fish, insects and plants (Haering and Habermann, 2021; Hao et al., 2021; Zhou et al., 2024). A true differential gene expression analysis requires an estimate of variance to perform hypothesis testing. The fuzzy-c means method used to produce clusters in Mfuzz, assigns genes to clusters and assigns a membership value between 0 and 1 for each gene. Genes with low membership influence the clustering results less than higher membership scores, meaning the method is less sensitive to noise than the traditional hard clustering methods (see Materials and Methods for details). The clusters confirmed that the most dynamic transcriptional shift occurred during the transitions between stages 12 and 13 for both species (Fig. 4 and Fig. S9). We then assessed whether the genes expressed in each cluster were significantly enriched for any Biological Process Gene Ontology (BP GO) terms using the Search Tool for the Retrieval of Interacting Genes/Proteins (String) (Szklarczyk et al., 2023) (Fig. S9). Clusters 18 and 1 in *M. abdita* and clusters 2 and 28 in *D. melanogaster* were significantly enriched for embryonic, body plan and systems development terms (Fig. 4A). Genes found in *M. abdita* cluster 19 and 3, and *D. melanogaster* clusters 9 and 12 exhibited a pronounced increase in expression at stage 13, and were enriched for terms related to the nervous system and cuticle development, respectively (Fig. 4B). Signal [$\log_{10}$(observed/expected)], strength [weighted harmonic mean between observed/expected ratio and -log(FDR)] and false discovery rates for each GO term listed in Fig. 4 can be found in Table S3. In addition to these similarities, the more extreme expression changes in *D. melanogaster* between stages 12 and 13 (Fig. 4A-C) could reflect the fact that *Drosophila* employs a specialized tissue (amnioserosa) and evolutionarily novel mechanisms for dorsal closure (Schmidt-Ott and Kwan, 2022).

### Investigation of 'orphan' genes exclusively expressed during embryogenesis in *M. abdita* reveals a novel F-Box LRR gene

As the evolution of new genes within lineages and species is an important means of diversification of developmental mechanisms (Chen et al., 2010, 2013), we searched the genome of *M. abdita* for 'orphan' genes. Orphan genes are either highly diverged from

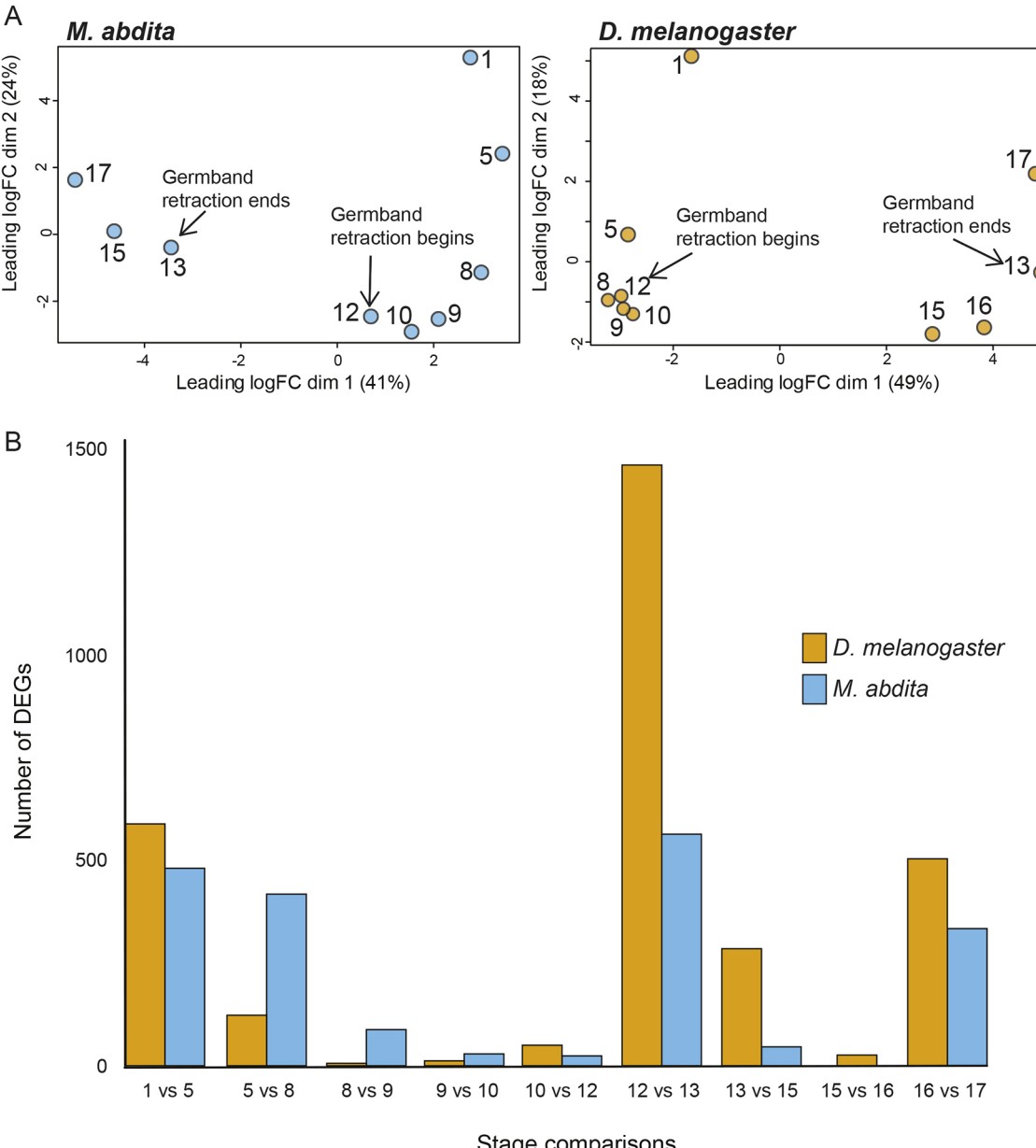

**Fig. 3. Multidimensional scaling, clustering and differential gene expression across developmental stages in *M. abdita* and *D. melanogaster*.** (A) Multidimensional scaling (MDS) plots based on the top 500 most differentially expressed genes from single-embryo RNA-Seq samples. The *M. abdita* data are shown on the left (blue) and the *D. melanogaster* data on the right (yellow). Points represent individual samples, and stages are labeled 1-17. Arrows indicate the stages when germband retraction begins and ends. (B) Number of differentially expressed genes (DEGs) between sequential developmental stages. in *M. abdita* and *D. melanogaster*.

known sequences or represent newly evolved genes (Vakirlis et al., 2020; Xia et al., 2025). In the genome annotation of *M. abdita*, eggNOG-mapper was unable to assign orthologs to 1049 gene models. Further searches using *blastn* on the coding sequences and *blastp* on the predicted protein sequences did not reveal any sequence similarity to other dipterans. Approximately 8.7% of the genes of *M. abdita* were 'orphan' genes.

We found that ~10% (109/1049) of these orphan genes were expressed during embryogenesis, including 24 that were exclusively expressed during embryogenesis (Fig. S10). Most of these genes exhibited sharp expression peaks, with 14 peaking at stage 5 (cellularization), 4 at stage 12 (germband retraction) and smaller groups peaking at stages 13, 15 and 17. Four genes showed

broader expression patterns, with two highly expressed during stages 8-10 (germband elongation) and two across stages 5-12 (cellularization to germband retraction). To confirm some of these transcriptomic patterns, we performed fluorescent *in situ* hybridization chain reaction (HCR) on three of the largest orphan genes. The HCR results corroborated the RNA-seq profiles: consistent with the sharp stage-specific RNA-seq peak of Scaffold_3.980 (Fig. 5A), Scaffold_3.980 is ubiquitously and exclusively expressed during cellularization (Fig. 5C), Scaffold_3.555 and Scaffold_1.5029 also showed expression patterns matching RNA-seq dynamics (Fig. 5A), transitioning from broad to ventral domains during gastrulation and germband extension (Fig. 5D,E).

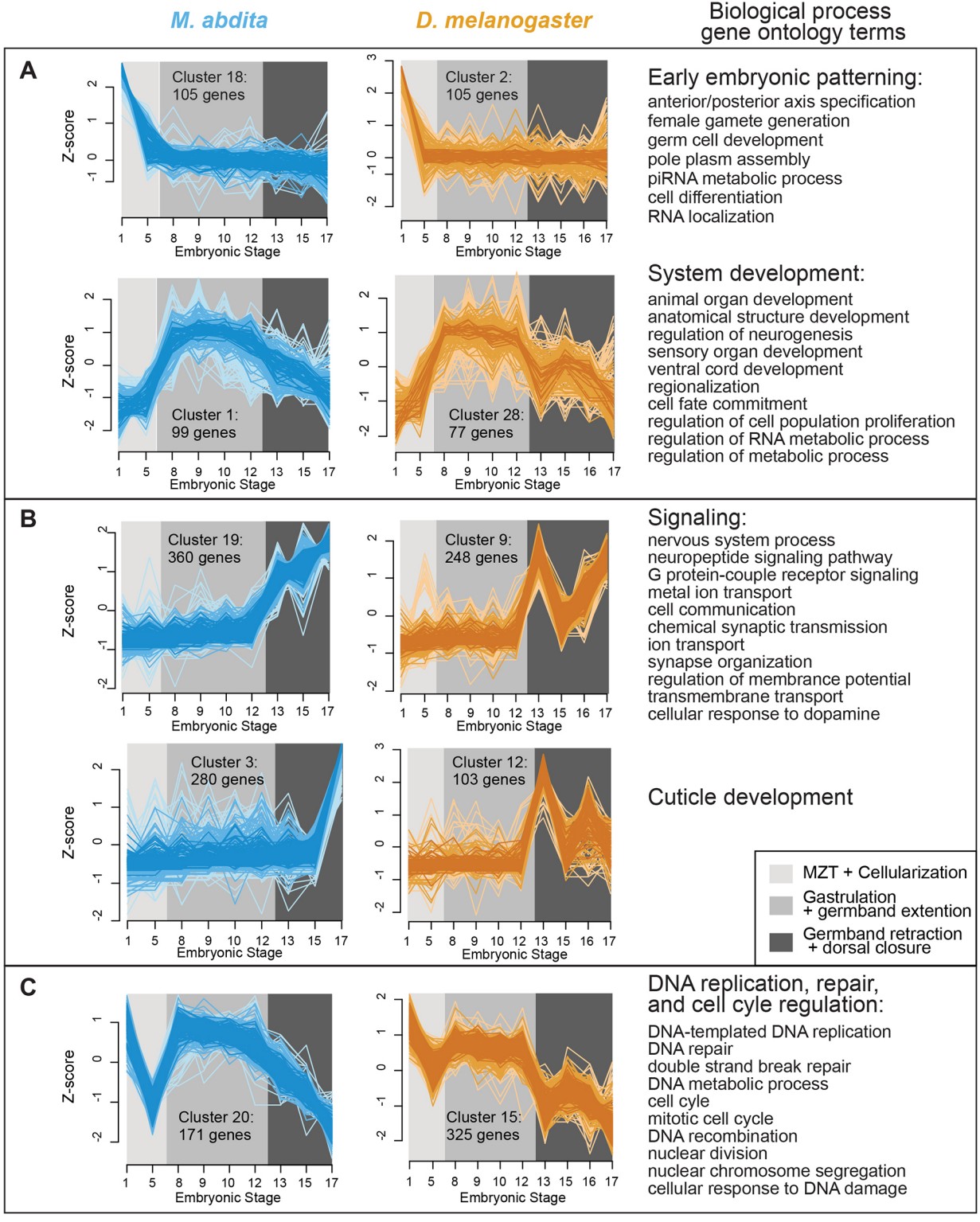

**Fig. 4. Expression dynamics of gene clusters and enriched GO terms during embryogenesis.** (A-C) Expression profiles of differentially expressed gene clusters during embryogenesis and their enriched Biological Process Gene Ontology (BP GO) terms for *M. abdita* (blue) and *D. melanogaster* (yellow). Each line is the expression profile of an individual gene with membership score above 0.7 (on a 0-1 scale) for that cluster. The darker the color, the higher the membership score of the gene for the cluster. The Mfuzz generated cluster names (e.g. 'Cluster 4') are arbitrary but retained here for continuity. The number of genes in each cluster is indicated below the cluster name. Maternal-zygotic transition (MZT)+cellularization, gastrulation+germband extension, and germband retraction+dorsal closure are highlighted in light gray, gray and dark gray, respectively. Shared enriched BP GO terms for the boxed clusters are listed on the right.

To further characterize these 24 embryo-specific orphan genes, we examined their open reading frames and protein sequences (Table S4). We classified 21 genes as likely to be coding genes and of those we further classified 9 as likely to encode stable proteins (blue highlighted rows in Table S4). Additional searches of InterProScan's database revealed no known protein domains or

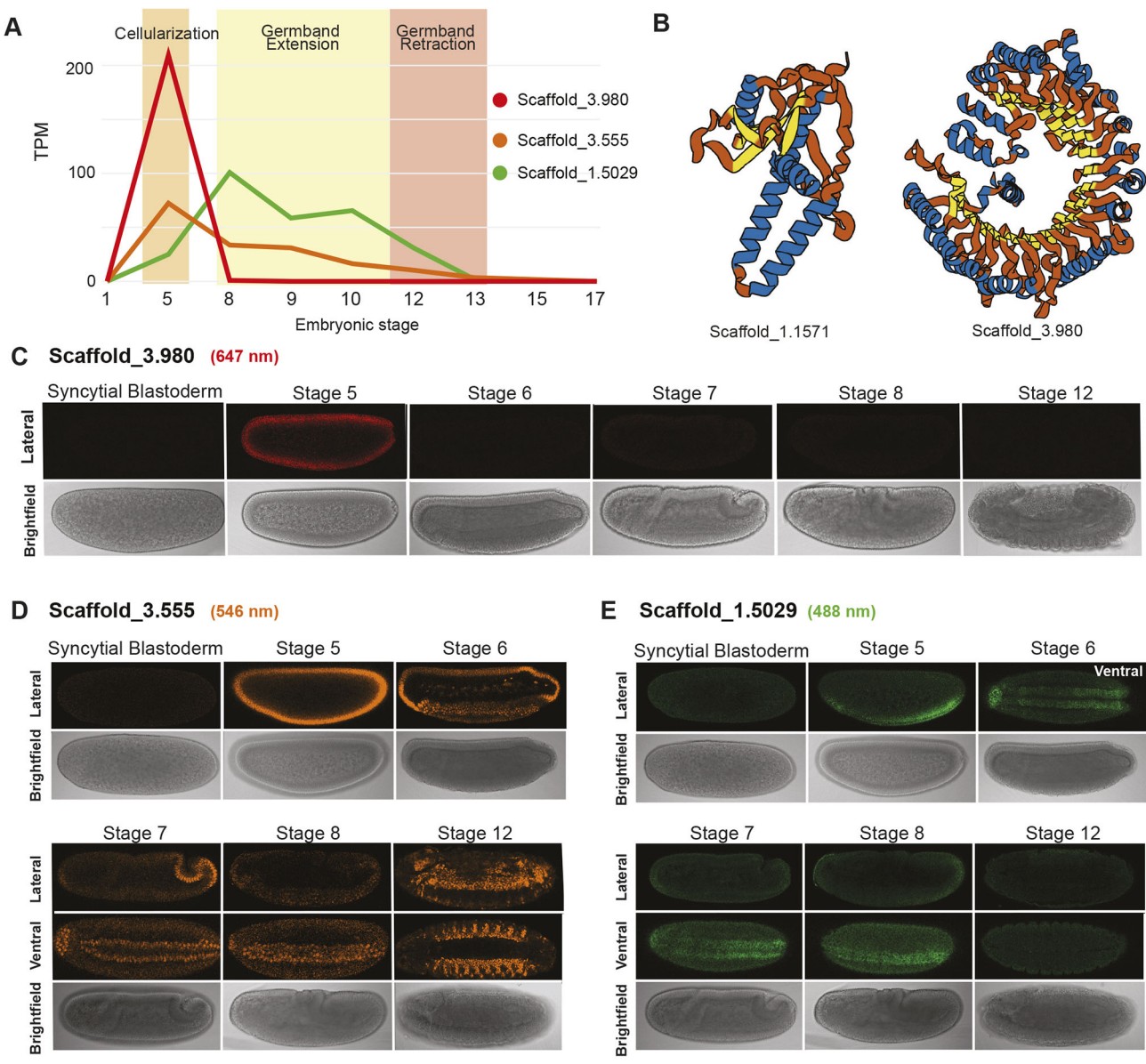

**Fig. 5. Select *M. abdita* orphan gene expression data and predicted protein structures.** (A) RNA-Seq expression profiles of three *M. abdita* orphan genes (Scaffold_3.980, Scaffold_3.555 and Scaffold_1.5029) during embryonic development. Developmental stages are indicated on the *x*-axis. Expression is measured in transcripts per million (TPM). (B) AlphaFold-predicted protein structures for Scaffold_1.1571 and for Scaffold_3.980. Structural similarity searches suggest that Scaffold_3.980 encodes a F-box-LRR protein. Alpha helices are highlighted in blue; β sheets are in yellow. (C-E) RNA-FISH HCR images for three orphan genes: Scaffold_3.980 (F-box LRR), Scaffold_3.555 and Scaffold_1.5029. Shown are representative single *z*-planes acquired at 20× magnification; anterior is left and dorsal is upwards in lateral view. Stages correspond to syncytial blastoderm, stage 5 (cellularization), stages 6 and 7 (gastrulation), stage 8 (germband extension), and stage 12 (germband retraction). Lateral bright-field images are included to aid staging and orientation.

any domains consistent with known transposases (Jones et al., 2014). We then used Alphafold2 to generate protein structure models (Jumper et al., 2021). Two of these genes resulted in confident structures (Table S4, pTM>0.5). Scaffold_1.1571 could not be related to any known protein (Fig. 5B); however, Scaffold_3.980 showed significant structural similarity to F-box leucine-rich repeat (LRR) proteins (Fig. 5B). As mentioned above, Scaffold_3.980 shows expression exclusively during cellularization (Fig. 5A) and expression dynamics were confirmed with FISH HCR (Fig. 5C).

We then compared the F-box LRR orphan protein sequence to known F-box LRR genes in *D. melanogaster* but found no obvious ortholog. We identified orthologs to *D. melanogaster* F-box LRR genes *Ppa*, *Kdm2*, *Fbxl4*, *Fbxl7*, *Fbl6*, *CG32085*, *CG9003* and *CG8272*. In total, we identified 15 F-box LRR genes in *M. abdita* (8 with clear *D. melanogaster* orthologs, 6 with orthologs in other dipteran species and our orphan gene). One of the dipteran F-box LRR orthologs found in *M. abdita* (Scaffold_3.3336) had an identical expression profile (stage 5) to the orphan F-box LRR gene (Scaffold_3.980).

F-box LRRs are components of the 'E3 ubiquitin ligase SCF complex' that ubiquitinates targeted proteins for later degradation by the cell. Specifically, the F-box domain binds to *Skp1* and the LRR domain binds to the target, i.e. the molecule identified for ubiquitination and degradation. F-box LRRs are known to ubiquitinate important developmental signaling molecules in

*D. melanogaster*, including the pair-rule gene *paired* ( *prd*), which is bound by the F-box LRR protein *Partner of paired* (*Ppa*) (Raj et al., 2000). *Ppa* is unusual in that its expression is patterned rather than uniform, as most F-Box LRR genes seem to be in *D. melanogaster* embryogenesis (Das et al., 2002). Given that, in *D. melanogaster*, the 12 best-known F-box LRR proteins (*Skp2*, *Ppa*, *Kdm2*, *FipoQ*, *Fbxl4*, *Fbxl7*, *Fbl6*, *CG32085*, *CG13766*, *CG12402*, *CG9003* and *CG8272*) are expressed throughout embryogenesis according to both our RNA-Seq data and ENCODE gene expression data, the stage-restricted expression of the orphan F-box LRR gene in *M. abdita* and 13 other orphan genes with expression peaking before gastrulation might reflect previously overlooked developmental differences between *M. abdita* and *D. melanogaster* at the blastoderm stage.

## Conclusions

The scuttle fly *M. abdita* is an important non-traditional model organism with hitherto very limited genomic resources. We have filled this gap by providing genomic and transcriptomic resources. By assembling a chromosome-level genome and annotating it, we revealed substantial chromosomal synteny with *D. melanogaster* while uncovering many orphan genes. Additionally, our comparative transcriptome analysis across embryogenesis highlights conserved and divergent regulatory dynamics. The discovery of a suite of orphan genes with exclusively pre-gastrular zygotic expression, including a novel F-box LRR gene, underscores that orphan genes may help to identify windows of accelerated evolution in developmental gene networks. Ultimately, we hope the addition of these resources will further comparative research of developmental mechanisms in Diptera and continue the development of Diptera as a model taxon more broadly.

## MATERIALS AND METHODS
### Generation of inbred *M. abdita* line

To generate single crosses of *M. abdita*, we collected pupae at the end of the pupal stage, when pupae darken approximately 1-2 days before eclosion, and transferred them to 35×10 mm Petri dishes (Fisher Scientific, 50-820-644) until hatching. We monitored the plates every morning and every 2 h during the day to collect virgin females. We paired each virgin female with a single male fly and allowed them to mate for 2 days in 35×10 mm Petri dishes containing a gel solution made from 2% agar in water. On the third day, we prepared egg-laying vials by boiling 0.8 g of a fish food mixture – composed of one part spirulina flakes (Aquatic Eco-Systems, ZSF5) and two parts sinking powder (Aquatic Eco-Systems, F1A) – in 10 ml of a 0.8% agar solution (EMD Millipore, 1.01614). We added approximately 1 ml of this solution to 10×75 mm culture tubes (Fisher Scientific,14-961-25) and allowed it to solidify. After cooling, we added 0.1 g of fish food on top using weighing paper, followed by 200 µl of water. We used a cotton swab to compact the food and clean any excess moisture from the sides of the tubes. We then transferred the mating pairs from the agar plates to the culture tubes and plugged the tubes with rayon balls (TIDI, 969162). See Fig. S11 for visual diagram of this process. We established multiple single crosses and tracked them using a progressive hierarchical code to identify the lineage. We conducted nine generations of sibling×sibling single-pair matings across three separate parallel lineages (A, B and C). At generation six, we generated pooled crosses within each lineage to allow the mixing of potentially lethal recessive alleles that may have accumulated during the single-cross procedure. Following this, we maintained the B lineage, as it exhibited higher overall fertility and health.

### Genome assembly
#### *De novo* library preparation, sequencing and assembly

We generated a *de novo* reference genome for *M. abdita* with sequencing data from HiFi PacBio reads and Dovetail's OmniC libraries (Cantata Bio). For HiFi PacBio sequencing, we collected and snap-froze in liquid nitrogen ~600 dechorionated embryos (mostly stages 16 and 17) from a 10-generation inbred line of a previously established laboratory culture of *M. abdita* Schmitz, 1959 (Schmidt-Ott et al., 1994). We sent these samples to Dovetail Genomics for HiFi PacBio library preparation and sequencing. Library preparation circularizes fragments so they can be read many times to generate a high-fidelity consensus sequence. Sequencing on the SMRT (single molecule, real-time) nanofluidic chip generates the long reads inherent to PacBio. PacBio generated 25.4 gigabases of paired reads (42× coverage) from which Dovetail generated a haplotype-resolved draft assembly using the Hifiasm assembler (Hifiasm v0.15.4-r347 with default parameters) (Cheng et al., 2021). We used blobtools v1.1.1 to identify possible contamination and removed 33 scaffolds from the assembly (Challis et al., 2020; Laetsch and Blaxter, 2017). After filtering out haplotigs and contig overlaps with purge_dups v1.2.5, 89 scaffolds remained (Guan et al., 2020).

#### Omni-C library preparation and sequencing

We collected and snap-froze an additional ~600-700 dechorionated embryos and ~60 young pupae in liquid nitrogen from the same inbred line of *M. abdita* for Omni-C Library preparation and sequencing. Omni-C is a proprietary technology for long-range proximity ligation and sequencing of genomic libraries that captures spatial information within the genome through chromatin fixation and sequencing. Omni-C differs from other Hi-C preparations in that the chromatin is digested with a sequence-independent endonuclease. This eliminates biases inherent to competitive restriction enzyme-based approaches. Samples were treated with formaldehyde to fix the chromatin and then digested with DNAse I. The resulting ends were then repaired, and biotinylated bridge adapters were ligated to the ends. Subsequent steps involved proximity ligation of adapter-ligated ends, reversal of formaldehyde-induced crosslinks and DNA purification. Non-internally ligated biotin residues were removed from the purified DNA. Sequencing libraries were prepared with NEBNext Ultra enzymes and Illumina-compatible adapters. Before PCR enrichment, biotin-containing fragments were isolated using streptavidin beads. Sequencing was performed on the Illumina HiSeqX platform.

#### Scaffolding assembly with HiRise

Both the Hifiasm draft assembly and Dovetail OmniC sequencing reads were used as input for HiRise, a software tailored for scaffolding genome assemblies with proximity ligation data (Putnam et al., 2016). Based on spatial data from the chromatin conformation capture (Fig. S1), HiRise pinpoints regions where contigs are joined incorrectly (misjoins) in the initial assembly and utilizes this spatial information to re-orient contigs and construct larger scaffolds. The OmniC library sequences were aligned to the draft assembly using the Burrows-Wheeler Aligner (Li and Durbin, 2009). HiRise-analyzed read pairs are mapped to the draft scaffolds to develop a genomic distance likelihood model. This model is then used to identify and break potential misjoins, score prospective joins and execute joins surpassing a set threshold. These scaffolds consist of sequentially arranged contigs separated by gaps. We used QUAST to calculate %GC, N50, L50 and Ns per 100 kb (Gurevich et al., 2013). We then repeat-masked the genome with RepeatMasker (Open-3.0. 1996-2010; http://www.repeatmasker.org). To estimate heterozygosity and sequencing error rates, we calculated the frequency spectrum of canonical 21-mers using jellyfish, and input the resulting histograms into GenomeScope (Marçais and Kingsford, 2011; Vurture et al., 2017). Using NUCmer (mummer v3.23 software package), we aligned the non-chromosome size scaffolds (4–15) back to the three chromosome-size scaffolds (1-3) and found repetitive sequences present in scaffolds 4-15 (Kurtz et al., 2004; Marçais et al., 2018).

### Genome annotation
#### Repeat masking the genome

To generate a custom library for repeat masking, we ran RepeatModeler v2.0.4 (Open-1.0. 2008-2015; https://github.com/Dfam-consortium/RepeatModeler) on the genome to find and/or model potential repeats. We then used RepeatMasker's script fambd.py v0.4.3 to extract Arthropoda records from the dfam database and combined these sequences with the RepeatModeler output to use as the repeat library. We used RepeatMasker v4.1.5 (http://www.repeatmasker.org) to soft-mask the genome with our custom library of repetitive low-complexity DNA and transposable elements.

### Data used as evidence of genome features

We downloaded available Illumina RNA-Seq data for *M. abdita* from NCBI which included paired-end reads from three adults and pooled embryos (Table S5). We generated RNA-Seq data for nine precisely staged embryos, first and third instar larval stages, and a 1-day-old pupal stage (Table S5). For protein evidence, we downloaded all Dipteran protein sequences from NCBI's RefSeq database, which included 2,122,027 sequences from 617 species (Sayers et al., 2021). We also downloaded Uniprot's complete protein sequence file (UniProt Release 2023_04), which contains 570,157 sequences from 14,509 species (The UniProt Consortium, 2023).

### Annotation pipeline

We annotated the reference genome of *M. abdita* using evidence from all RNA-Seq transcripts (Table S5), protein sequences and gene prediction software (Fig. S3). The choices of software used in this pipeline are based on the methods section of VanKuren's 'Draft *Papilio alphenor* assembly and annotation' (VanKuren, 2023). First, we assembled RNA transcripts using two transcript assemblers: Stringtie v2.2.1 and Trinity v2.15.1 (Pertea et al., 2015; Grabherr et al., 2011; Haas et al., 2013). Trinity performs both genome-guided and *de novo* assembly; we assembled transcripts using both methods. For Trinity and Stringtie's genome-guided assemblies, we first mapped reads to the genome with 'Spliced Transcripts Alignment to a Reference' software (STAR v2.7.10b) (Dobin et al., 2013). Next, we used the 'Program to Assemble Spliced Alignments' (PASA v2.5.3) to identify gene structures from all three assemblies (Haas et al., 2003). We predicted gene models directly from the PASA assemblies using the PASA plug-in TransDecoder v5.7.1, which identifies candidate coding regions from Trinity and StringTie assemblies. To create protein alignments, we used the software Exonerate v2.2.0, which maps protein sequences to the genome (Slater and Birney, 2005). We used the BRAKER3 pipeline (braker.pl v3.0.6) to predict gene models from mapped RNA-Seq reads and protein data (Gabriel et al., 2024). BRAKER3 relies on the software Augustus and GeneMark to predict gene models (Stanke et al., 2006; Brůna et al., 2020). We also generated our own *ab initio* gene structure predictions, using GlimmerHMM v3.0.4 (Majoros et al., 2004). First, we collected "hints" for training the *ab initio* predictors by extracting protein-coding hints from the protein alignments using Augustus' exonerate2hints function, intron hints from mapped RNA-Seq reads using Augustus' bam2hints function, and exon/intron hints from the PASA assemblies using Augustus' bam2exonhints function. We additionally used the coding predictions from Transdecoder to create training models and further refined those models using the lib.selectTrainingModels function from Funannotate (Palmer and Stajich, 2019). This training data was used to run GlimmerHMM with hints as guidance (Majoros et al., 2004). We then provided the PASA assemblies, mapped protein data, TransDecoder predictions, *ab initio* predictions, BRAKER3 predictions and a file weighting each line of evidence to the software Evidence Modeler v2.1.0. Evidence Modeler constructed the consensus gene structures that were updated by PASA to add UTRs and identify alternative transcripts (Haas et al., 2008). We removed gene structures that overlapped with RepeatMasker output using a Funannotate function called "RemoveBadModels." We then used Funannotate v1.8.1 to identify annotations that match known transposable elements (TEs) and repeat proteins using BLAST and updated the annotation to remove remaining TEs. We used AGAT v1.4.1 to remove genes with an open reading frame (ORF)<100 amino acids in length and any associated gene structures (Dainat, 2022). We assigned gene names using eggNOG-mapper v2.1.12 which relies on orthology predictions to functionally annotate genes (Cantalapiedra et al., 2021; Huerta-Cepas et al., 2019).

### Synteny analysis

We used MCScanX (primary release) to compare the synteny of *M. abdita* and *D. melanogaster* genomes (Wang et al., 2012). MCScanX identifies syntenic blocks based on a score given to each gene pair. We set these as follows: match_score=50 (default), match size=5 (number of genes required to constitute a syntenic block), gap_penalty=0 (no penalty for gaps) and max_gaps=100. We used the output of this run to generate our synteny map. We used the circlize R package to plot the results (Gu et al., 2014).

### Embryo staging for RNA sequencing (RNA-seq)

For precise embryo staging, embryos of the appropriate age were mounted on a microscope slide under halocarbon 27 oil and observed in a Zeiss Axiophot compound microscope equipped with a 10× objective and DIC (differential interference contrast) optics until they reached the desired stage. They were photographed and immediately processed for RNA extraction as previously described (Lott et al., 2014). We collected *M. abdita* and stage-matched *D. melanogaster* embryos at stages 1, 5, 8, 9, 10, 12, 13, 15, 16 and 17. Photos of the sequenced embryos can be found in Fig. S5.

### RNA isolation and sequencing

We incubated each sample for 5-10 min at room temperature in TRIzol. We then froze each sample in TRIzol at −80° C. We extracted total RNA using the TRIzol/phenol-chloroform protocol detailed in the appendix (protocol 10) of 'Functional evolution of a morphogenetic gradient' by Chun Wai Kwan (Kwan, 2017). The University of Chicago genomics core facility constructed cDNA libraries using the TruSeq kit with PCR (Illumina). The cDNA libraries were barcoded and multiplexed for 100 bp paired-end sequencing on one lane of a HiSeq Illumina 2000 sequencer.

### Differential gene expression analysis

*M. abdita* had ~2× the reads for each sample compared to *D. melanogaster* (Fig. S6A). We chose not to down sample *M. abdita* reads to match *D. melanogaster* to avoid losing the power to detect changes among genes across *M. abdita* developmental stages. We justified this by looking for any evidence that the higher number of reads skewed the relationship between gene expression and variance, but found that the relationship between mean expression and variance in expression is similar in both species without subsetting the data (Fig. S7). Additionally, the average, median and variance in gene expression (TPM) were very similar across development in both species (Fig. S8).

We aligned *M. abdita* RNA-Seq reads to the reference genome generated in this publication and *D. melanogaster* RNA-Seq reads to *D. melanogaster*'s genome (Accession number GCF_000001215.4). We used Subread v2.0.5 to align reads (Liao et al., 2013). 90-99% of reads mapped to the genome for each sample. We input the aligned bam files into Subread's featureCount function, which assigns the reads to a genomic feature from the annotation file (gff). At this point, we performed the analysis in Rstudio (Posit Software; http://www.posit.co/) using the free and open-source statistical language R (https://www.R-project.org/) and various R packages, including EdgeR (Robinson et al., 2009), tidyr (v1.3.1; https://tidyr.tidyverse.org), dplyr (v1.1.4; https://dplyr.tidyverse.org) and ggplot2 (Wickham, 2016). We used EdgeR to assess gene expression differences between samples. We filtered out lowly expressed genes and normalized data by library size (TMM normalization) for both species. We calculated counts per million (CPM) to normalize the difference in raw reads between samples and then calculated transcripts per million (TPM) to account for differences in gene length. We calculated gene length as the coding sequence length for the longest isoform of each gene.

As our RNA-Seq samples do not include biological replicates, we used two different methods to analyze expression differences and patterns in our dataset: (1) a hard clustering method to estimate a global dispersion for the two RNA-Seq datasets in order to perform a traditional analysis with hypothesis testing to identify differentially expressed genes (DEGs); and (2) a soft clustering method to look at expression patterns in our time course dataset, which does not require an estimate of variance.

### Hard clustering and traditional hypothesis testing

To identify DEGs between stages, we estimated the squared coefficient of variation (BCV) using k-means clustering of the samples. We began by calculating a distance matrix for the samples and extracting the first four eigenvectors, which together explained >95% of the variance. To determine the optimal number of clusters (k), we calculated the within-cluster sum of squared errors (WSS) and Silhouette scores for k values ranging from 1 to 8, selecting k=5 based on these metrics. For *M. abdita*, clustering with k=5 grouped the embryonic stages as follows: group 1 (stage 1), group 2 (stage 5), group 3 (stages 8-12), group 4 (stages 13-15) and group 5 (stage 17). For *D. melanogaster*, the clusters were: group 1 (stage 1), group 2 (stages 5 and

8), group 3 (stages 9-12), group 4 (stages 13-16), and group 5 (stage 17). Using these groupings, we estimated the dispersion as σ=0.36 for both *M. abdita* and *D. melanogaster*. These estimates were applied globally. Differential expression analysis was performed using EdgeR's exactTest() function, comparing gene expression between pairwise embryonic stages rather than the k-means groups. Genes were considered differentially expressed if they met the criteria: fold change $\leq-3$ or $\geq3$, and $P<0.001$.

### Soft clustering to look at developmental expression patterns

To examine gene expression dynamics during embryonic development, we performed soft clustering using the R package Mfuzz on our RNA-Seq datasets (Kumar and Futschik, 2007). Mfuzz employs the fuzzy c-means algorithm, which assigns each gene a membership score between 0 and 1 for each cluster. Unlike traditional hard clustering, this approach allows genes to partially belong to multiple clusters. Genes with lower membership values contribute less to the definition of the cluster, making the method more robust to noise. Prior to clustering, Mfuzz standardizes expression data so that each gene profile has a mean of zero and a standard deviation of one (i.e. Z-score normalization). As a result, the *y*-axis of Mfuzz plots represents standardized expression values (Z-scores), where 0 corresponds to the gene's average expression across all time points, positive values reflect above-average expression and negative values indicate below-average expression.

### Gene ontology enrichment analysis

We used the STRING database v12.0 (Search Tool for the Retrieval of Interacting Genes/Proteins) to perform enrichment analysis of Biological Process GO terms for genes with membership scores at or above 0.7 for each cluster output by Mfuzz (Kumar and Futschik, 2007). STRING compares input gene sets to a reference genome to identify networks of interacting genes and enrichments in biological processes. Since *M. abdita* is not available in STRING, we first used NCBI's Blast tool to select the top *D. melanogaster* ortholog match or hit for each *M. abdita* gene (Altschul et al., 1990). Using FlyBase's batch download tool, we retrieved the corresponding FlyBase IDs, which STRING accepts as input (Öztürk-Çolak et al., 2024). STRING performed the enrichment analysis identifying the Biological Process GO terms significantly associated with each developmental stage. STRING measures enrichment based on the strength of enrichment [Log_{10}(observed/expected)], false discovery rate (*P*-values corrected for multiple testing with Benjamini-Hochberg) and the signal [weighted harmonic mean between observed/expected ratio and -log(FDR)].

### 'Orphan' gene identification

We identified genes from *M. abdita*'s annotation file for which eggNOG-mapper could not assign an ortholog. Next, we assessed the expression of these genes in our RNA-Seq data. To identify genes with exclusively embryonic expression, we focused on those with a CPM>1 in at least one of the nine embryonic stages and a CPM<1 in all pupal, larval and adult stages. We validated these genes further by analyzing their ORFs using NCBI's 'Open reading frame finder'. We used CPC2 to evaluate the nucleic acid sequences to assess coding potential (Kang et al., 2017). Finally, we performed sequence similarity searches using NCBI's blastn and blastp tools, querying both nucleotide and protein sequences against the entire NCBI database, as well as against Dipteran-specific sequences (Altschul et al., 1990; Sayers et al., 2021).

### Protein structure prediction and structural similarity search

We used Protparam to assess protein stability, aliphatic index and hydropathicity of predicted proteins (Gasteiger et al., 2005). We then searched all predicted protein sequences against InterProScan to look for any missed domains, specifically to look for evidence of transposable elements that were not discovered with blast searches (Jones et al., 2014). We used the AlphaFold2.ipynb provided by ColabFold v1.5.5 to predict protein structures (Jumper et al., 2021; Mirdita et al., 2022). If the predicted protein had a predicted template modeling (pTM) score>0.5 we then uploaded the structure to FoldSeek's website and searched the available databases (AlphaFold/Proteome, AlphaFold/Swiss-Prot, AlphaFold/UniProt50, BFMD) for proteins with similar structure (van Kempen et al.,

2024; Varadi et al., 2022, 2024). We used Protein Imager to generate publication-quality images of protein structures (Tomasello et al., 2020).

### Fluorescent *in situ* hybridization chain reaction and imaging

We designed probes for five *M. abdita* genes, each paired with distinct HCR hairpins (B1-B3; Molecular Instruments). Specifically, *Mab-zen* (*evm.model.Scaffold_2_212802314.3491*) was assigned to B2, *Mab-zen-like* (*evm.model.Scaffold_2_212802314.3495*) to B3, *evm.model.Scaffold_3_155577901.980* to B1, *evm.model.Scaffold_3_155577901.555* to B2, and *evm.model.Scaffold_1_222469999.5029* to B3. Probe sets were ordered as 50-pmol oPools oligos from IDT, and sequences are listed in Table S6. HCR staining followed the protocol of Henderson et al. (2024). Embryos were processed in two batches, each stained with two or three probes simultaneously. Batch 1 included *Mab-zen* (B2) and *Mab-zen-like* (B3). Batch 2 included *evm.model.Scaffold_3_155577901.980* (B1), *evm.model.Scaffold_3_155577901.555* (B2) and *evm.model.Scaffold_1_222469999.5029* (B3). Fluorophores were imaged in separate channels: B1 at 647 nm, B2 at 546 nm and B3 at 488 nm, with DAPI counterstaining imaged at 358 nm. Confocal imaging was performed at 20× magnification on a Zeiss LSM 900. For Batch 1, imaging parameters were standardized across embryos: B1 at 5% laser intensity, 30 μm pinhole, 620 V gain; B2 at 3.5%, 30 μm, 530 V; B3 at 6.5%, 30 μm, 620 V. For Batch 2, B1 was set at 3.5%, 30 μm, 640 V; B2 at 1%, 30 μm, 530 V; B3 at 2.2%, 30 μm, 580 V; and DAPI at 1.8%, 30 μm, 690 V. Within each batch, all embryos were imaged using identical settings. Notably, channels corresponding to the B3 hairpin consistently showed higher background or tissue autofluorescence than B1 and B2.

### Acknowledgements

We thank Lily Shiue and Qianyu Jin at Cantata Bio for assembling the genome. We thank Nicholas VanKuren for his notes on genome annotation. We thank Yaikhomba Mutum for the conversation on protein structure prediction and structural similarity searches. We thank Xiang-Ru (Shannon) Xu for optimization of the *M. abdita* single cross procedure and the drawings in Fig. S11. We thank the University of Chicago's Center for Research Informatics and Research Computing Center for hosting and maintaining the high-performance computing clusters used during this work. JETSTREAM2: This work used Jetstream2 at Indiana University through allocation BIO220075 from the Advanced Cyberinfrastructure Coordination Ecosystem: Services & Support (ACCESS) program, which is supported by National Science Foundation grants 2138259, 2138286, 2138307, 2137603 and 2138296. GBCF: Bioinformatic work was supported in part by the Notre Dame University Genomics and Bioinformatics Core Facility.

### Competing interests
The authors declare no competing or financial interests.

### Author contributions
Conceptualization: U.S.-O.; Data curation: A.T.-T., E.A., V.G., H.Y., S.A.S.; Formal analysis: A.T.-T.; Funding acquisition: U.S.-O.; Investigation: A.T.-T., E.A., V.G., C.W.K., U.S.-O.; Methodology: A.T.-T., C.W.K., H.Y., S.A.S.; Project administration: U.S.-O.; Resources: S.A.S., U.S.-O.; Software: H.Y., S.A.S.; Supervision: U.S.-O.; Visualization: A.T.-T.; Writing – original draft: A.T.-T., U.S.-O.; Writing – review & editing: A.T.-T., E.A., V.G., C.W.K., S.A.S.

### Funding
Research reported in this publication was supported by the National Institute of General Medical Sciences of the National Institutes of Health under Award Number R01GM127366 and by the University of Chicago. The funders had no role in study design, data collection and analysis, decision to publish, or preparation of the manuscript. Open Access funding provided by the University of Chicago. Deposited in PMC for immediate release.

### Data and resource availability
Genome: the annotated *Megaselia abdita* genome can be found in NCBI's Genome database under BioProject Accession PRJNA1164289 (GenBank Assembly Accession number: GCA_048544405.1). RNA-Seq data: all fastq files containing the raw sequencing reads for each sample have been uploaded to NCBI's Sequence Read Archive (SRA) and can be found under the BioProject Accession PRJNA1200075. Individual BioSample and SRA accession numbers can be found in Table S5, including those samples that were not generated in this study but used as evidence for the annotation of the genome. Genome Annotation Pipeline: We are developing a genome annotation pipeline in conjunction with the University of

Chicago's Research Computing Center (RCC). The GitHub repository can be found at https://github.com/himanshyadav/genomes and contains SLURM job scheduler scripts and configurations for running genome annotation software used in this study on HPC clusters. While specifically written for the University of Chicago RCC's Midway3 Cluster using a SLURM job scheduler, scripts can be easily modified and the repository will continue to be updated. To enhance reproducibility and portability, we are developing Nextflow-based workflows for genome annotation which will be added to the repository as they are finalized. Genome Browser: Jetstream2 at Indiana University is a resource provider for NSF's ACCESS program, which aims to broaden access to super computing resources at no cost to researchers. To access the *Megaselia abdita* genome browser and related tools, create an ACCESS ID, then use this ID, create an account and login to Jetstream2. You will apply for an 'allocation' of credits which can be used on Jetstream2. Detailed instructions on the use of Jetstream2 can be found at https://jetstream-cloud.org/get-started/index.html. Alternatively, a portable Docker image is available on GitHub (https://github.com/kallistaconsulting/genomic_resources_megaselia), providing a reproducible, ready-to-use environment that can be deployed across platforms without additional installation steps. All other relevant data and details of resources can be found within the article and its supplementary information.

### Peer review history
The peer review history is available online at https://journals.biologists.com/dev/lookup/doi/10.1242/dev.204732.reviewer-comments.pdf

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
