## [Peer Review File · Development (Cambridge, England)]

Genomic resources for the scuttle fly *Megaselia abdita*: a model organism for comparative developmental studies in flies

Ayse Tenger-Trolander, Ezra Amiri, Valentino Gantz, Chun Wai Kwan, Himanshi Yadav, Sheri A. Sanders and Urs Schmidt-Ott

DOI: 10.1242/dev.204732

Editor: Cassandra Extavour

Review timeline

Original submission:	19 February 2025
Editorial decision:	2 April 2025
First revision received:	10 September 2025
Accepted:	6 October 2025

Original submission

First decision letter

MS ID#: dev.204732

MS Title: Genomic Resources for the Scuttle Fly *Megaselia abdita*: A Model Organism for Comparative Developmental Studies in Flies

Authors: Ayse Tenger-Trolander; Ezra Amiri; Valentino Gantz; Chun Wai Kwan; Sheri A. Sanders; Urs Schmidt-Ott

Article Type: Techniques and Resources Article

Dear Dr Tenger-Trolander,

I have now received all the referees' reports on the above manuscript, and have reached a decision. The referees' comments are appended below, or you can access them online: please go to:

Please carefully consider especially the comments of Reviewer 1, as without replicates, which I understand may be technically infeasible for a revision, I agree with this Reviewer that many of the analyses need to account for the lack of replicates, and the interpretation of the results should be tempered accordingly.

As you will see, the referees express considerable interest in your work, but have some significant criticisms and recommend a substantial revision of your manuscript before we can consider publication. If you are able to revise the manuscript along the lines suggested, which may involve further experiments, I will be happy receive a revised version of the manuscript. Your revised paper will be re-reviewed by one or more of the original referees, and acceptance of your manuscript will depend on your addressing satisfactorily the reviewers' major concerns. Please also note that Development will normally permit only one round of major revision. If it would be helpful, you are welcome to contact us to discuss your revision in greater detail. Please send us a point-by-point response indicating your plans for addressing the referees' comments, and we will look over this and provide further guidance.

Please attend to all of the reviewers' comments and ensure that you clearly highlight all changes made in the revised manuscript. Please avoid using 'Tracked changes' in Word files as these are lost in PDF conversion. I should be grateful if you would also provide a point-by-point response detailing

how you have dealt with the points raised by the reviewers in the 'Response to Reviewers' box. If you do not agree with any of their criticisms or suggestions please explain clearly why this is so.

Reviewer 1

In this manuscript, Tenger-Trolander et al., report the genome assembly of *Megaselia abdita* and a series of transcriptomes along its embryogenesis that authors use to study gene expression dynamics together with equivalent *D. melanogaster* RNA-seq datasets. The assembled genome of *M. abdita* consist on 3 major scaffolds which corresponded to the 3 chromosomes previously described for this species. The authors also investigate synteny with *D. melanogaster* and find high correspondence of syntenic blocks between these two dipteran species. Moreover, they observe a shift of gene expression during germ band retraction. Finally, the authors identify almost 30 orphan genes, including a F-box LRR gene, in the genome of *M. abdita* expressed during its embryogenesis.

Although this work provides a series of resources of great interest for the scientific community interested in insect and Diptera genomics and evolution, I have some concerns/comments for the authors that should be addressed before publication:

1. My main concern regarding this work is the lack of replicates for the RNA-seq data and the conclusions made from the analyses on them. While I acknowledge the efforts that authors made in order to circumvent this caveat, I find the analyses are not entire appropriate to reach any conclusion. If I understood correctly, they clustered their samples based on BCV, -which in any case gave them some clusters made of individual samples-, but later they didn't use these clusters and run the DEG analysis between sequential developmental stages. Therefore, this analysis with no replicas seems a bit poor to state the conclusions authors made.

Given this lack of replicas, if authors are not able to repeat the analyses, which I'd totally understand because it will imply to repeat all the RNA extractions and analysis to avoid batch effects, they may prefer to use alternative approaches to study the dynamics of gene expression in their datasets. My recommendation would be to try for instance Mfuzz, which does not require biological replicas, since this software is precisely used to analyse dynamics of gene expression and it has been broadly utilised in previous works for insects and beyond: Pallarès-Albanell <https://doi.org/10.1242/dev.203017>; Zhou [10.1186/s12870-024-04731-3](https://doi.org/10.1186/s12870-024-04731-3) ; Almudi <https://doi.org/10.1038/s41467-020-16284-8>; Ioannidis <https://doi.org/10.1186/s12864-021-08274-x> ; Maeso <https://doi.org/10.1186/s12915-016-0267-0>; Marlétaz <https://doi.org/10.1038/s41586-018-0734-6>; Cardoso-Moreira <https://doi.org/10.1016/j.celrep.2020.108308>. By performing this soft clustering, authors will be able to address these dynamic patterns of expression and to do the subsequent GO term enrichment analysis using the genes that are contained in these clusters.

2. In the same manner, the results regarding the expression of individual genes along embryogenesis: zen and zen-like and the orphan genes seem not strongly supported by this lack of replicas. Perhaps in these cases authors could use an alternative approach and perform qPCRs with several replicates for some of these genes to strengthen these results and conclusions.

3. Minor: It would be useful to indicate the number of genes in each of the clusters in figure 4 and S9 or in the new clustering plots if authors decide to perform Mfuzz or another alternative analysis.

Reviewer 2

SUMMARY OF THE ADVANCE MADE IN THIS PAPER AND ITS POTENTIAL SIGNIFICANCE TO THE FIELD

Tenger-Trolander and coauthors submitted the manuscript titled "Genomic Resources for the Scuttle Fly *Megaselia abdita*: A Model Organism for Comparative Developmental Studies in Flies" for consideration for Development. This manuscript was evaluated as a "Techniques and Resources Article". The authors point out how insects, including the order Diptera offer a great animal system to study evolution and development through the comparative approach. The genus *Drosophila* is robust with genomic and transcriptome resources and many other Dipteran groups have some resources too, out to the more distantly-related mosquitoes. A taxon that has not yet gotten its omics due is the family Phoridae. This family diverged from the lineage of *Drosophila* around 154

million years ago. Among Phoridae is the species *Megaselia (M.) abdita*, or the scuttle fly, which has been used for decades to study embryo development and its evolution.

This manuscript shares the group's construction and characterization of a quality *M. abdita* genome and genome browser replete with analysis tools. The authors provide some interesting details of the genome, including the character of the Hox cluster, breadth of repetitive sequences, synteny with the *Drosophila (D.) melanogaster* genome, and orphan genes. The authors also performed and characterized comparative transcriptomes of *M. abdita* and *D. melanogaster* from numerous successive embryo stages. While many of the chronicled omics observations lacked a known or suspected developmental or evolutionary importance, the manuscript certainly delivers for techniques and resources. I recommend this manuscript for publication and provide some minor revisions to be made below.

SUGGESTIONS TO AUTHORS

Minor revisions:

Abstract line 31. Change "One of them," to "One of these dipteran species is"

Results and discussion lines 103-105 was vague about the outcome. Please clarify.

Line 112, change "UTRs" to "untranslated regions (UTRs)"

Line 113 change "Eggnog-mapper" to "eggNOG-mapper"

Line 183, Figure S6B. I think it is worth saying something about the surprising finding of the substantial increase in expressed genes at the later development stages than the earlier ones.

Lines 224 and 229 and perhaps elsewhere. I suggest putting *Drosophila* in italics.

Lines 259 and 272. I suggest putting the CG genes in italics too.

Line 368 should soft-masked be soft-mask?

Figure 1. I suggest including NCBI and DTL as 0 for Platypezoidea. Putting the zeros in bold red color to draw attention to the absence of resources here. The red arrow that exists draws attention to Platypezoidea but not the absence of genomic resources.

Table 1. Misspelled Eukaryotic as Eukaroyotic

Table 2. Change "mrnas" to "mRNAs"

Figure 2B. Put Zerknüllt and Zerknüllt-like in red color. Also, genes names should be in italics.

Figure 3A needs a key for the color circles. Need to make this figure more user friendly.

Figure 4. (A) misspelled differentiation

Figure 4. (B) misspelled development

Figure S2. What is coverage on the x-axis? Clarify this.

First revision

Author response to reviewers' comments

Dear Editors and Reviewers:

Thank you both for your thoughtful and constructive feedback on our manuscript. We followed the Reviewers suggestion to use an alternative approach such as Mfuzz to better capture gene expression dynamics. We appreciate this suggestion and have implemented Mfuzz clustering. The results are consistent with our previous analysis using DEGs and provide a reassuring second line of evidence. Reviewer 1 also encouraged us to validate individual gene expression patterns, which motivated us to add new RNA-FISH HCR experiments that corroborate our RNA-Seq data and provide additional spatial information. Thank you as well for pointing out several areas where improved presentation would increase clarity. We have carefully addressed these points, including clarifying results descriptions, correcting terminology, and formatting, and improving figures and tables. We are grateful for the reviewers' careful reading and comments, which have helped us improve the clarity, rigor, and impact of the manuscript. Below we separately address each point of the reviewers and point out how we addressed them.

In addition to changes made based on reviewer comments, we would like to highlight a few updates. First, we have added the GitHub link for the Genome Browser Docker image, allowing users to run the browser locally or on their own servers, rather than relying solely on NSF ACCESS

and Jetstream2. Second, we have initiated a collaboration with the University's Research Computing Center to develop a genome annotation pipeline in NextFlow. This pipeline is available on GitHub (link provided in manuscript) and will continue to be updated as the resource develops. As a result of this development, we have added a co-author, Himanshi Yadav.

Comments from the Reviewers in Black, Responses in Blue:

Reviewer 1: In this manuscript, Tenger-Trolander et al., report the genome assembly of *Megaselia abdita* and a series of transcriptomes along its embryogenesis that authors use to study gene expression dynamics together with equivalent *D. melanogaster* RNA-Seq datasets. The assembled genome of *M. abdita* consist of 3 major scaffolds which corresponded to the 3 chromosomes previously described for this species. The authors also investigate synteny with *D. melanogaster* and find high correspondence of syntenic blocks between these two dipteran species. Moreover, they observe a shift of gene expression during germ band retraction. Finally, the authors identify almost 30 orphan genes, including a F-box LRR gene, in the genome of *M. abdita* expressed during its embryogenesis.

Although this work provides a series of resources of great interest for the scientific community interested in insect and Diptera genomics and evolution, I have some concerns/comments for the authors that should be addressed before publication:

1. My main concern regarding this work is the lack of replicates for the RNA-Seq data and the conclusions made from the analyses on them. While I acknowledge the efforts that authors made in order to circumvent this caveat, I find the analyses are not entire appropriate to reach any conclusion. If I understood correctly, they clustered their samples based on BCV, -which in any case gave them some clusters made of individual samples-, but later they didn't use these clusters and run the DEG analysis between sequential developmental stages. Therefore, this analysis with no replicas seems a bit poor to state the conclusions authors made.

Given this lack of replicas, if authors are not able to repeat the analyses, which I'd totally understand because it will imply to repeat all the RNA extractions and analysis to avoid batch effects, they may prefer to use alternative approaches to study the dynamics of gene expression in their datasets. My recommendation would be to try for instance Mfuzz, which does not require biological replicas, since this software is precisely used to analyse dynamics of gene expression and it has been broadly utilised in previous works for insects and beyond: Pallarès-Albanell <https://doi.org/10.1242/dev.203017>; Zhou [10.1186/s12870-024-04731-3](https://doi.org/10.1186/s12870-024-04731-3) ; Almudi <https://doi.org/10.1038/s41467-020-16284-8>; Ioannidis <https://doi.org/10.1186/s12864-021-08274-x> ; Maeso <https://doi.org/10.1186/s12915-016-0267-0>; Marlétaz <https://doi.org/10.1038/s41586-018-0734-6>; Cardoso-Moreira <https://doi.org/10.1016/j.celrep.2020.108308>. By performing this soft clustering, authors will be able to address these dynamic patterns of expression and to do the subsequent GO term enrichment analysis using the genes that are contained in these clusters.

We appreciate the reviewer's thoughtful feedback regarding the limitations of differential expression analysis without biological replicates. We have now implemented the Mfuzz soft clustering approach to analyze dynamic gene expression across developmental time points, as recommended. This clustering approach, which is specifically designed for time series data without replicates, allowed us to identify gene expression trajectories and perform downstream GO term enrichment analysis based on cluster membership. These new results are presented in the revised manuscript in both the results (lines 227-236) and methods sections (lines 513-524) as well as in Figures 4 and S9.

Regarding the reviewer's concern about our use of hard clustering, we would like to clarify that this analysis was performed solely to estimate a reasonable global dispersion value for downstream differential expression analysis. In the absence of replicates, commonly used RNA-Seq tools suggest using a fixed dispersion value (e.g., ~0.1 for inbred model organisms, ~0.4 for human samples). To refine this estimate for our dataset, we grouped samples using BCV-based clustering and derived a dispersion estimate from that grouping, which we then applied globally. We have clarified this point in the manuscript in the results (lines 200-207) and methods (lines 497-511). We then proceed with a classical pairwise DEG analysis for each stage using the estimated dispersion. Using a

relatively high estimate (0.36) provides a conservative framework, meaning that only genes with strong differences across stages are identified as differentially expressed.

The differential expression analysis included in the manuscript was intended to illustrate the relative magnitude of transcriptional change across stages, particularly the marked increase in differentially expressed genes between stage 12 (germ band retraction) and stage 13 (beginning of dorsal closure), which we observed in both species. We find this coincidence informative but acknowledge the limitations of DEG analysis without replicates and are open to removing this section entirely if the reviewers strongly feel that it detracts from the study.

2. In the same manner, the results regarding the expression of individual genes along embryogenesis: *zen* and *zen-like* and the orphan genes seem not strongly supported by this lack of replicates. Perhaps in these cases authors could use an alternative approach and perform qPCRs with several replicates for some of these genes to strengthen these results and conclusions.

To address this concern, we decided to pursue RNA-FISH HCR to assess the quality of the RNA-Seq data and to add some spatial information on expression. We stained for *Mab-zen* and *Mab-zen-like* in a developmental time series and have added the results to Figure S4B as well as to the results section (lines 174-175). The expression is consistent with the RNAseq data.

We also used RNA-FISH HCR to assess the RNA-Seq data for three of the largest orphan genes and again found the images consistent with the RNA-Seq expression data. See Figure 5C, D, and E and lines 264-270 in the results section.

Descriptions of the HCR probes, protocol, and imaging are included in the methods section: 'Fluorescent *in situ* hybridization chain reaction and imaging' starting on line 565.

3. Minor: It would be useful to indicate the number of genes in each of the clusters in figure 4 and S9 or in the new clustering plots if authors decide to perform Mfuzz or another alternative analysis.

We have added the number of genes in each mfuzz generated cluster with a membership score > 0.7.

Reviewer 2: SUMMARY OF THE ADVANCE MADE IN THIS PAPER AND ITS POTENTIAL SIGNIFICANCE TO THE FIELD

Tenger-Trolander and coauthors submitted the manuscript titled "Genomic Resources for the Scuttle Fly *Megaselia abdita*: A Model Organism for Comparative Developmental Studies in Flies" for consideration for Development. This manuscript was evaluated as a "Techniques and Resources Article". The authors point out how insects, including the order Diptera offer a great animal system to study evolution and development through the comparative approach. The genus *Drosophila* is robust with genomic and transcriptome resources and many other Dipteran groups have some resources too, out to the more distantly-related mosquitoes. A taxon that has not yet gotten its omics due is the family Phoridae. This family diverged from the lineage of *Drosophila* around 154 million years ago. Among Phoridae is the species *Megaselia (M.) abdita*, or the scuttle fly, which has been used for decades to study embryo development and its evolution.

This manuscript shares the group's construction and characterization of a quality *M. abdita* genome and genome browser replete with analysis tools. The authors provide some interesting details of the genome, including the character of the Hox cluster, breadth of repetitive sequences, synteny with the *Drosophila (D.) melanogaster* genome, and orphan genes. The authors also performed and characterized comparative transcriptomes of *M. abdita* and *D. melanogaster* from numerous successive embryo stages. While many of the chronicled omics observations lacked a known or suspected developmental or evolutionary importance, the manuscript certainly delivers for techniques and resources. I recommend this manuscript for publication and provide some minor revisions to be made below.

SUGGESTIONS TO AUTHORS

Minor revisions:

Abstract line 31. Change "One of them," to "One of these dipteran species is"
 Changed. See line 34.

Results and discussion lines 103-105 was vague about the outcome. Please clarify.
 We have revised this section to improve clarity. See lines 106-108.

Line 112, change "UTRs" to "untranslated regions (UTRs)"
 Corrected. See line 115.

Line 113 change "Eggnog-mapper" to "eggNOG-mapper"
 Corrected. All occurrences of *eggnog-mapper* changed to *eggNOG-mapper*.

Line 183, Figure S6B. I think it is worth saying something about the surprising finding of the substantial increase in expressed genes at the later development stages than the earlier ones. We did not find this result to be surprising since this is bulk RNA-Seq of individual embryos rather than single cell RNA-Seq data. As the genome activates around stage 5, more genes are expressed and as tissues differentiate during development the total number of genes expressed should grow during embryogenesis even if individual tissues may express fewer genes. But perhaps it is surprising that there is notable increase in both species between stages 12 and 13. The shift from stage 12 to stage 13 also has significant changes in differential gene expression which we discuss in the section 'Major transition in transcriptional expression profile during germband retraction'.

Lines 224 and 229 and perhaps elsewhere. I suggest putting *Drosophila* in italics.
 All instances of *Drosophila* are now italicized.

Lines 259 and 272. I suggest putting the CG genes in italics too.
 Corrected. See lines 284 and 297.

Line 368 should soft-masked be soft-mask?
 It should. Corrected. See line 393.

Figure 1. I suggest including NCBI and DTL as 0 for Platyzoidea. Putting the zeros in bold red color to draw attention to the absence of resources here. The red arrow that exists draws attention to Platyzoidea but not the absence of genomic resources.
 Changed.

Table 1. Misspelled Eukaryotic as Eukaroyotic
 Corrected.

Table 2. Change "mrnas" to "mRNAs"
 Corrected.

Figure 2B. Put Zerknüllt and Zerknüllt-like in red color. Also, genes names should be in italics.
 Corrected.

Figure 3A needs a key for the color circles. Need to make this figure more user friendly.
 We have removed these colors.

Figure 4. (A) misspelled differentiation
 Corrected.

Figure 4. (B) misspelled development
 Corrected.

Figure S2. What is coverage on the x-axis? Clarify this.
 We have added two sentences to the figure caption (S2A) explaining coverage and frequency in a k-mer distribution. It now reads:

“A) 21-mer frequency distribution for *M. abdita* PacBio reads. The observed k-mer coverage (blue) is modeled by GenomeScope (black line), which includes contributions from unique sequences (yellow) and sequencing errors (orange). In this context, ‘coverage’ on the x-axis refers to the number of times a k-mer is observed in the sequencing reads and ‘frequency’ on the y-axis to the number of unique k-mers with that number of observations. In our data, we see $\sim 4.5 \times 10^7$ unique 21-mers (frequency) that were each observed ~ 20 times (coverage).”

Second decision letter

MS ID#: dev.204732R1

MS Title: Genomic Resources for the Scuttle Fly *Megaselia abdita*: A Model Organism for Comparative Developmental Studies in Flies

Authors: Ayse Tenger-Trolander; Ezra Amiri; Valentino Gantz; Chun Wai Kwan; Himanshi Yadav; Sheri A. Sanders; Urs Schmidt-Ott

Article Type: Techniques and Resources Article

Dear Dr Tenger-Trolander,

I am happy to tell you that your manuscript has been accepted for publication in Development, pending our standard publication integrity checks.

Reviewer 1

The revised version of Tenger-Trolander et al. manuscript has improved significantly. They have fully addressed all my comments as well as those from the other reviewer. I think it is now ready for publication as a resource. I only have 4 very minor edits:

1. Could you include the p-values for the GO enrichment for each of the species? If it is too messy to include them in the main figure because there are two sps, at least, it would be good to have them in the suppl. Figure or as a table, where each species GO and Mfuzz are separated.
2. Line 197: figure 4B should be S4B?
3. Lines 278-285: This section needs some references, such as some previous works using this method to help the justification of its use and/or the one that implemented Mfuzz itself
4. Line 407: "our" instead of "one"

Reviewer 2

SUMMARY OF THE ADVANCE MADE IN THIS PAPER AND ITS POTENTIAL SIGNIFICANCE TO THE FIELD

This revised manuscript shares the group's construction and characterization of a quality *M. abdita* genome and genome browser replete with analysis tools. The authors provided some interesting details of the genome, including the character of the Hox cluster, breadth of repetitive sequences, synteny with the *Drosophila (D.) melanogaster* genome, and orphan genes. The authors also performed and characterized comparative transcriptomes of *M. abdita* and *D. melanogaster* from numerous successive embryo stages. While many of the chronicled omics observations lacked a known or suspected developmental or evolutionary importance, the manuscript certainly delivers for techniques and resources. The authors took the reviewers comments and critiques seriously and revised the original manuscript into a more polished product that is ready to be published. I recommend this revised manuscript for publication.

Second revision

Author response to reviewers' comments

1. Could you include the p-values for the GO enrichment for each of the species? If it is too messy to include them in the main figure because there are two sps, at least, it would be good to have them in the suppl. Figure or as a table, where each species GO and Mfuzz are separated.

STRING measures enrichment based on the strength of enrichment ($\text{Log}_{10}(\text{observed}/\text{expected})$), false discovery rate (p-values corrected for multiple testing with Benjamini-Hochberg), and the signal (weighted harmonic mean between observed/expected ratio and $-\log(\text{FDR})$).

We have included a new table (S3) in the supplement with signal, FDR, and strength for the clusters GO terms listed in figure 4 for both species.

2. Line 197: figure 4B should be S4B?

Corrected.

3. Lines 278-285: This section needs some references, such as some previous works using this method to help the justification of its use and/or the one that implemented Mfuzz itself

We have added a new sentence with references at line 229:

Mfuzz has been widely applied for developmental time-series RNA-seq analyses across diverse systems, including fish, insects, and plants (Haering, M. and Habermann, 2021; Hao et al., 2021; Zhou et al., 2024).

4. Line 407: "our" instead of "one"

I did not see a usage of 'one' that should be replaced with 'our.'